# Cambrian Chordates and Vetulicolians

Mark A. S. McMenamin 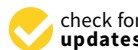

Department of Geology and Geography, Mount Holyoke College, South Hadley, MA 01075, USA;
mmcmenam@mtholyoke.edu; Tel.: +1-413-538-2280

**Abstract:** Deuterostomes make a sudden appearance in the fossil record during the early Cambrian. Two bilaterian groups, the chordates and the vetulicolians, are of particular interest for understanding early deuterostome evolution, and the main objective of this review is to examine the Cambrian diversity of these two deuterostome groups. The subject is of particular interest because of the link to vertebrates, and because of the enigmatic nature of vetulicolians. Lagerstätten in China and elsewhere have dramatically improved our understanding of the range of variation in these ancient animals. Cephalochordate and vertebrate body plans are well established at least by Cambrian Series 2. Taken together, roughly a dozen chordate genera and fifteen vetulicolian genera document part of the explosive radiation of deuterostomes at the base of the Cambrian. The advent of deuterostomes near the Cambrian boundary involved both a reversal of gut polarity and potentially a two-sided retinoic acid gradient, with a gradient discontinuity at the midpoint of the organism that is reflected in the sharp division of vetulicolians into anterior and posterior sections. A new vetulicolian (*Shenzianyuloma yunnanense* nov. gen. nov. sp.) with a laterally flattened, polygonal anterior section provides significant new data regarding vetulicolians. Its unsegmented posterior region ('tail') bears a notochord and a gut trace with diverticula, both surrounded by myotome cones.

**Keywords:** chordates; vetulicolians; cephalochordates; vertebrates; tunicates; Cambrian; *Banffia*; *Vetulicolia*; *Myllokunmingia*; *Metaspriggina*; *Shenzianyuloma*; agnathans; notochord; gut diverticula; myomeres; deuterostomes; all-*trans*-retinoic acid (ATRA); segmentation clock; morphogen gradients

---

## 1. Introduction

In his 1888 book *The Geological Evidences of Evolution*, Angelo Heilprin, Curator of Invertebrate Paleontology and the Philadelphia Academy of Natural Sciences, made the following prediction [1]:

> "*Looking at the animal kingdom broadly … we find that of the two great divisions into which that kingdom is divided, the backboned or vertebrate animals, like the fish, reptile, amphibian, and quadruped, and those without backbone, the Invertebrata, like the coral, starfish, crab, etc., only the latter is represented in the earliest period, the Cambrian, in which indisputable animals remains have been found. Not a vestige of any of the higher forms has here been met with. But let me warn you against this non-appearance. It is by no means impossible, or indeed unlikely, that backboned animals already lived during this period of time, and that their remains will still someday be discovered.*"

Heilprin's prediction [1] concerning the discovery of Cambrian vertebrates was vindicated one hundred and eleven years later with the description of *Myllokunmingia* in 1999.

In a debate over the number of phyla in the Cambrian, S. J. Gould [2] said of the enigmatic Burgess Shale form *Banffia* (classified here as a vetulicolian):

> "*Consider Banffia, namesake of the more famous national park adjoining Yoho and the Burgess Shale. Walcott's 'worm' … is almost surely a weird wonder.*"

Gould was correct to consider *Banffia* a weird wonder, in other words, a member of a previously unknown phylum. This conclusion was problematic for Burgess Shale researchers such as Derek Briggs and Richard Fortey, who believe that there were not nearly as many Cambrian phyla [3] as inferred by Gould [2]. Gould's interpretation, however, gained additional credence with recognition of the early Cambrian phylum Vetulicolia [4]. Phylum Vetulicolia, plus newly recognized early Cambrian chordates, lend weight to the concept of a sudden, rather than muted and long-drawn-out, Cambrian Explosion [5–7].

## 2. Chinese Breakthrough

Two developments in paleontology were to have great impact on both the study of Proterozoic-Cambrian paleobiology and Cambrian chordates. Continuing early Western interest in Chinese Proterozoic and Cambrian strata and fossils [8,9], modern paleontological analysis of the boundary interval began in the late 1970s and early 1980s [10–12]. A critical subsequent event was the discovery in the mid-1980s of the Chengjiang Lagerstätte of the Yunnan Province, China, dated to Cambrian Series 2 [13].

Apart from *Pikaia*, which had long been known from the Burgess Shale (but not recognized as a chordate until 1979), the soft-bodied preservation of the Chengjiang fossils, colorfully preserved in iron oxides, has provided an astonishing glimpse at the soft-bodied early Cambrian fauna, such as the giant sclerite-lacking lobopodian *Paucipodia inermis*. Other examples of exceptional fossil preservation include five genera of scleroctenophores (skeletonized comb jellies) recently described from Chengjiang [14].

Discoveries from the Wulongqing Formation (Guanshan Biota; Cambrian Series 2, Stage 4) of Yunnan, China, have amplified this signal [15], as have even more recent discoveries from the Qingjiang biota [16]. Included in this suite of fossils are the oldest known chordates.

## 3. Cambrian Chordates

A major impediment to the study of Cambrian chordates (including the phyla/subphyla Cephalochordata, Urochordata [tunicates] and Vertebrata) is that, for many forms, the most taxonomically diagnostic features (with the exceptions, fortunately, of myomeres and the notochord) tend to be the ones most prone to rapid decay, thus leading to the potential absence of key apomorphies. This results in 'stemward slippage,' in other words, a fossil specimen appearing more primitive than it actually is because the key derived characters are not preserved [17]. Described or reported genera of Cambrian chordates are as follows.

### 3.1. Cathaymyrus

Assigned to the Cephalochordata [18,19], *Cathaymyrus* includes two, possibly synonymous, species, *C. diadexus* and *C. haikoensis*. Some researchers consider the genus to be "a chordate of uncertain affinity" [13], with D. Shu calling it "an amphioxus-like creature" [4]. The animal is remarkable for its elongate, narrowly tapered body divided by S-shaped myomeres. A notochord and a discontinuous gut trace may be present [20]. The animal may possess a notochord but lacks a well-defined cranial region. *Cathaymyrus* is possibly synonymous with *Zhongxiniscus* [13].

### 3.2. Cheungkongella

*Cheungkongella* is an attached, spindle-shaped sessile suspension-feeding animal that is thought to be either an early tunicate [21] or a cambroernid (an extinct non-chordate deuterostome clade that includes forms such as the eldoniids, *Phlogites* and *Herpetogaster* [22]). D. Shu [4] considers *Cheungkongella* to be the oldest known urochordate.

### 3.3. Haikouichthys

This agnathan fish (Figure 1) was described from the *Eoredlichia* Zone, Qiongzhusi Formation near Ercaicun, Yunnan, China [23]. Sense organs and associated structures tentatively identified in this chordate include eyes, nasal sacs, and otic capsules [4]. Possibly synonymous with *Myllokunmingia* [13].

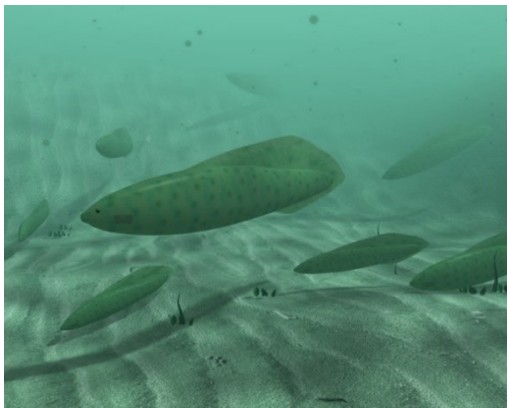

**Figure 1.** *Haikouichthys ercaicunensis*. Animals are approximately 2.5 cm in length. Image credit: Talifero. Used here per CC BY-SA 3.0.

### 3.4. Haikouella

This early chordate and possible agnathan [24] is represented by two species, *Haikouella lanceolata* and *H. jianshanensis* [25]. Filamentous gills are present that are similar to those of *Yunnanozoon*. In some specimens, the gill filaments coming from the gill structure appear to be paired. Fin rays are present in *Haikouella*, as are W-shaped (zigzag) myomeres. Tiny (100 micron) structures that are likely pharyngeal teeth are among the first such structures recorded in the deuterostome fossil record. The presence of these teeth serve to distinguish *Haikouella* as an agnathan chordate as opposed to the otherwise similar Cambrian hemichordate *Yunnanozoon* [24–29]. Interestingly, *Haikouella* is interpreted to have had both a ventral and a dorsal nerve chord [4].

### 3.5. Metaspriggina

The first appearance of gill bar structures (pharyngeal gills) occurs in the Middle Cambrian fossil from the Burgess Shale *Metaspriggina walcotti* (Figure 2). Initially thought to be related to the Ediacaran ecdysozoan genus *Spriggina*, *Metaspriggina* is now known to be a notochord-bearing chordate with features such as eyes and pharyngeal bars that are comparable to those of fish [30,31]. In addition, the myomeres of *Metaspriggina* have a distinct dorsal curve and W-shaped pattern formed by an added chevron in the ventral region. Although possibly due to homoplasy, this myomere configuration strongly resembles the fish pattern, suggesting that *Metaspriggina* might very well be ancestral to the gnathostomatans.

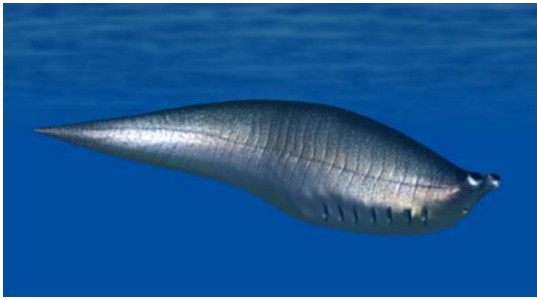

**Figure 2.** *Metaspriggina walcotti*. Length of animal 10 cm. PaleoArt credit: Nobu Tamura. Used here per CC BY-SA 4.0.

### 3.6. Myllokunmingia

The most ancient putative vertebrate is *Myllokunmingia fengjiaoa* from the Lower Cambrian *Eoredlichia* Zone, Qiongzhusi Formation near Ercaicun, Yunnan, China [13,19,32]. Its assignment to the vertebrates is supported by its filamentous gills, dorsal fin, W-shaped myomeres and especially, by what appear to be paired sense organs in the head region. Fin rays are present in *Myllokunmingia*, as are W-shaped (zigzag) myomeres. *Myllokunmingia* is possibly synonymous with *Haikouichthys* [13].

### 3.7. Pikaia

*Pikaia gracilens* (Figure 3) was initially described as a polychaete worm [33,34] in early descriptions of soft-bodied forms from the Burgess Shale (Middle Cambrian, *Bathyuriscus-Elrathina* Zone). *Pikaia* eventually became famously known as the earliest described Cambrian chordate [2]. In a definitive breakthrough that initiated the study of Cambrian chordates, Simon Conway Morris, in 1979, noted the similarity of *Pikaia* to the living cephalochordates *Amphioxus* and *Branchiostoma*, and thus extended the geological range of Chordata back to the Middle Cambrian [35].

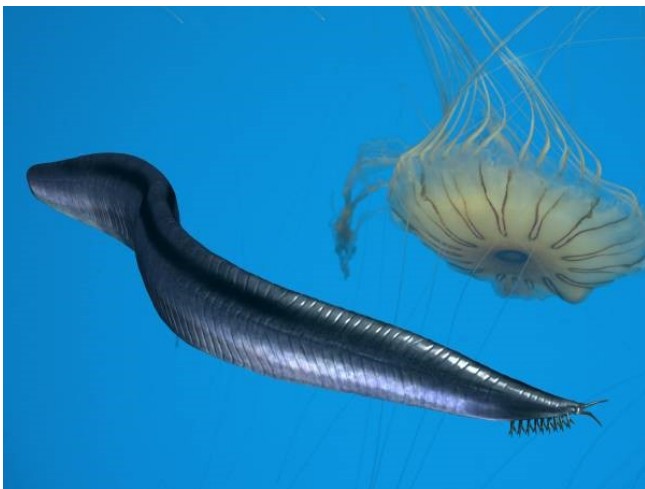

**Figure 3.** *Pikaia gracilens.* Length of animal 4 cm. Jellyfish in the background is a scyphozoan. PaleoArt credit: Nobu Tamura. Used here per CC-BY SA 4.0.

The previous most ancient chordate fossils were latest Cambrian to Cambro-Ordovician dermal plates of 'ostracoderm' fishes such as *Anatolepis*, *Sacabambaspis*, *Arandaspis* and *Astraspis*, the latter best known from the Middle Ordovician Harding Sandstone, Cannon City, Colorado, USA [36–38].

Recent analyses of *Pikaia* have placed it as a basal chordate [39,40], and indeed chordate affinity seems plausible but other possibilities, such as assignment to a new higher taxon, cannot be dismissed considering that *Pikaia* may have formed a cuticle (as occurs in some nonvertebrate cephalochordates) [41].

Large specimens of *Pikaia* reach 4 cm in length. Originally thought [35] to bear a notochord, the structure running along the animal's back is now called the 'dorsal organ', and a notochord and/or notochord plus nerve cord may occur in a position ventral to the dorsal organ [42]. True notochords of course do occur in modern *Amphioxus* and *Branchiostoma*. Unique features of *Pikaia* include short appendage-like structures possibly attached to the gill slits, and hagfish-like tentacles at the anterior end of the animal.

An analysis of swim mechanics in *Pikaia* concluded that it must have been a slow swimmer because it lacked the fast-twitch fibers that allow rapid motion in modern fish and other living chordates [42].

### 3.8. Shankouclava

Known only from the Lower Cambrian Maotianshan Shale at Anning, Kunming, South China, *Shankouclava anningense* (Figure 4) is widely regarded as the oldest known tunicate, thanks to its apparent similarity to an individual zooid of the living, colonial aplousobranch tunicate *Clavelina* [43]. A stolon-bearing, club-shaped animal, *Shankouclava* reaches 4 cm in length. In addition to its branchial basket, a possible endostyle occurs on the fossil.

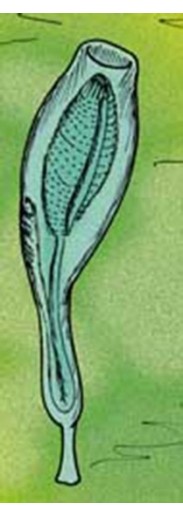

**Figure 4.** Cambrian tunicate *Shankouclava anningense*. The animal is 4 cm in height. Image credit: Apokryltaros. Used here per CC BY-SA 3.0.

### 3.9. Undescribed Shankouclava-like Form

A new fossil tunicate strongly resembling *Shankouclava*, but bearing tentacles, awaits description [13].

### 3.10. Yunnanozoon

The debate regarding the chordate [27,28], hemichordate [29] or even vetulicolian affinities of *Yunnanozoon lividum* (Figure 5) has underscored the challenges in interpreting this early chordate-like animal. One of the most strikingly beautiful members of the Chengjiang biota (occurring at both Chengjiang area and in Haikou), *Yunnanozoon*, has been variously assigned to the cephalochordates, stem-bilaterians, stem-deuterostomes, stem-ambulacrarians, stem-chordates, stem-hemichordates, crown hemichordates, cristozoans, vetulicolians and craniates, with one group of researchers concluding that it is best considered at present "as a bilaterian of uncertain affinity" [13].

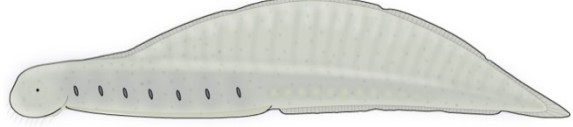

**Figure 5.** *Yunnanozoon lividum*. Animal is 6 cm in length. PaleoArt credit: Cyrus Theedishman. Used here per CC BY-SA 3.0.

Known from hundreds of specimens ranging from 2–6 cm in length, *Yunnanozoon* differs from the otherwise comparable *Haikouella* by having pharyngeal teeth that are an order of magnitude larger (1 mm) and yet, ironically, *Yunnanozoon* has a stomach that is not as large. Seriation occurs in *Yunnanozoon* as indicated by multiple (13!) paired gonads. As in *Haikouella lanceolata*, in some specimens, the gill filaments coming from the gill arch appear to be paired. Although there exists no clear evidence for a notochord nor myomeres in *Yunnanozoon*, filamentous structures attached to

body-axis-parallel rods occur on both sides of the anterior end of the animal. It seems evident that the filamentous arches functioned as gills.

### 3.11. Zhongjianichthys

This slender eel-like creature, 1.1 cm in length, has a thick integument [26]. No myomeres are visible in the fossil but these may be masked by its thick skin in the unique holotype. Thought by some [13] to be a preservational variant of *Myllokunmingia*, *Zhongjianichthys* is considered here to be a valid, distinct genus, although possibly synonymous with *Cathaymyrus* [13]. If myomeres are indeed absent, there may be a parallel between *Zhongjianichthys* and a genetically altered zebrafish (*Danio*) that shows normal development of fast muscle striations but never develops myotomes due to gene knockouts (tbx6⁻, her1⁻, her7⁻) that stop the segmentation clock [44]. These knockouts also lead to deformed vertebrae (scoliosis, etc.) but the mutant fish is nevertheless able to swim, the biomechanical demands being less in an aquatic environment than for tetrapods on land. *Zhongjianichthys* may represent an early chordate with an altered segmentation clock and no myotomes, but nevertheless, with an ability to swim. Putative anterior sensory structures include eyes, nasal sacs and a curious 'anterior-dorsal lobe' [4] that may have performed a sensory function.

### 3.12. Zhongxiniscus

*Zhongxiniscus* (Figure 6) is a slender cephalochordate-like animal known from Haikou, Kunming, China [20]. Although *Zhongxiniscus* has along its dorsal margin, triangular fins, the genus appears to lack both fin rays and W-shaped myomeres, developing instead S-shaped, narrow (140–145 microns width) myomeres as in *Cathaymyrus*.

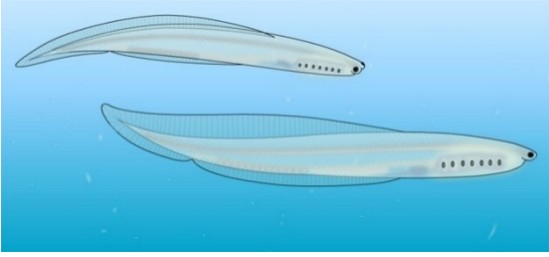

**Figure 6.** *Zhongxiniscus*. Length of animal 1 cm. PaleoArt credit: Cyrus Theedishman. Used here per CC BY-SA 3.0.

## 4. Vetulicolians

These enigmatic creatures, typified by the eponymous genus *Vetulicola*, have the general appearance of a segmented tadpole. They were originally identified as arthropods due to the apparently segmented character of the 'tail' in some genera, but the complete absence of jointed appendages plus the recognition of a series of five pairs of gill structures [45] in the anterior part of the animal led to a revised placement with the deuterostomes. They are best known from *Lagerstätten* such as Chengjiang [13] and have recently been reported from the newly discovered Qingjiang biota [16].

The overall shape of vetulicolians is somewhat reminiscent of bivalved crustaceans, but any resemblance to anterior tagmosis of crustaceans is superficial. The lateral line in vetulicolians represents a potential plane of flexure and this may be a characteristic of their unusual body plan.

Diego García-Bellido and his coauthors [46] concluded that "the common ancestor of the Vetulicolia + Tunicata include distinct anterior and posterior body regions—the former being non-fusiform and used for filter feeding and the latter originally segmented—plus a terminal mouth, absence of pharyngeal bars, the notochord restricted to the posterior body region, and the gut extending to the end of the tail."

Two informal groups of vetulicolians are recognized here, the "soft shell vetulicolians" and the "hard shell vetulicolians". Soft shell vetulicolians lack segmentation in their posterior section or tail; lack a prominent constriction between the anterior and posterior portions of the body; and have a very

blunt, rectangular anterior. Hard shell vetulicolians have a segmented posterior section, and may have a tuberculate anterior section, at least as juveniles.

### 4.1. Banffia

*Banffia constricta* (Figure 7), the first described putative vetulicolian, is a common constituent of the Middle Cambrian Burgess Shale biota. *Banffia* can attain a length of 10 cm. Originally assigned to the annelids, a lack of appendages has proved to be an insurmountable obstacle for assigning *Banffia* to Arthropoda.

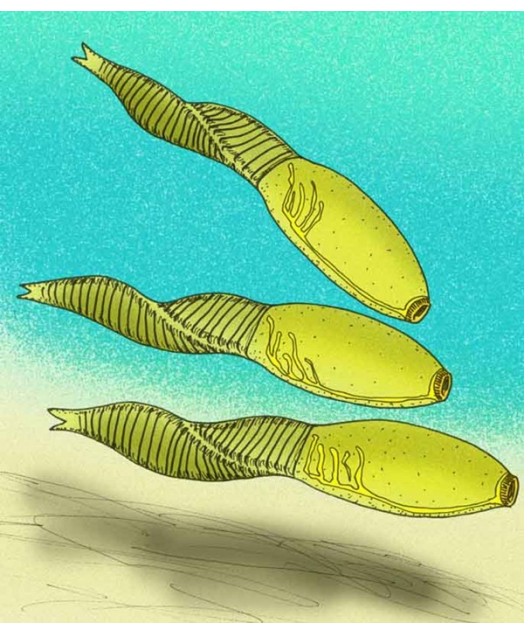

**Figure 7.** *Banffia constricta.* Length of animal 10 cm. PaleoArt credit: Apokryltaros. Used here per CC BY-SA 3.0.

As is characteristic for vetulicolians, *Banffia* is divided into an anterior part and a posterior part. The posterior has a characteristic twist and numerous (40–50) narrow segments. There is a "crossing pattern" twisted section at the constriction between the anterior and posterior sections. The anus is at the end of the posterior section and occurs in the center of a forked structure called the caudal notch [47]. The anterior section or 'carapace' is elongate pill-shaped and largely smooth. The anterior tip of the animal bears a corona-like structure of concentric rings or 'circlets'. The structure is presumed to be the mouth of the animal.

Fine anastomosing ridges on *Banffia*, identified as "micro-ichnofossils" [47], are in fact part of the structure of the animal and may represent vascular canals of some type. Comparable features occur on *Vetulicola longbaoshanensis* [15]. No 'gill slits' are present in *Banffia*, and this has impeded comparisons with other vetulicolians. J.-B. Caron [47] sees the "presence of mid-gut diverticulae in *Banffia*" to represent a possible link to protostomes.

*Banffia episoma* of Utah has a considerably diminished anterior section [48]. *Banffia confusa* from Chengjiang has gill slits on its anterior section and is thus securely placed in the Vetulicolia [47].

### 4.2. Beidazoon

*Beidazoon venustum* is considered a 'dwarf vetulicolian' reaching only 8–14 mm in length [49]. Tubercles occur on the anterior section. Thus, the possibility that the fossil may represent a juvenile cannot be excluded. *Beidazoon* is one of the 'hard shell' vetulicolians [49].

### 4.3. Bullivetula

*Bullivetula variola* is considered here to possibly be synonymous with (a juvenile of) *Beidazoon venustum*. Pronounced tubercles and pits occur on the anterior section [50].

### 4.4. Didazoon

Somewhat resembling *Pomatrum*, *Didazoon haoae* (Figure 8) has an anterior section divided into six segments, a posterior section divided into seven segments, and a creased junction between the two [51]. *Didazoon* is one of the 'soft shell' vetulicolians [49]. Five gill ports have been identified on either side of the anterior section. The tail section is advanced dorsally.

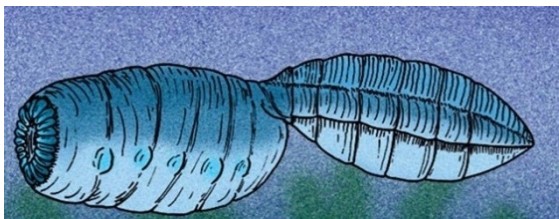

**Figure 8.** *Didazoon haoae*. PaleoArt credit: Stanton F. Fink. Used here per CC BY-SA 2.5.

### 4.5. Heteromorphus

*Heteromorphus confusus* was originally assigned to *Banffia*, and the possibility exists that more genera and species will need to be described with ongoing study of these morphologically variable fossils [50]. *Heteromorphus* is one of the 'soft shell' vetulicolians [49].

### 4.6. Heteromorphus Subtype New Species Form A

This undescribed form is a 'soft shell' vetulicolian with an apparently unsegmented posterior region ('tail') that is "covered with numerous wrinkles" [49]. The rather thick tail is thought to be either an extension of both the dorsal and ventral partitions of the anterior section, or, more probably, only an extension of the dorsal part of the anterior section [49].

### 4.7. Nesonektris

*Nesonektris aldridgei* (Figure 9) is a vetulicolian described in 2014 from the Cambrian Emu Bay Shale Konservat-Lagerstätte of Kangaroo Island, Australia. It has a rather rectangular anterior portion and a laterally flattened, paddle-like posterior section. Narrow, diamond-shaped intersegmental membranes occur between posterior section segments. Gill structures are not evident on the anterior section, but a lateral groove appears on both right and left sides of the anterior section along the line where one would expect to find gill structures [46].

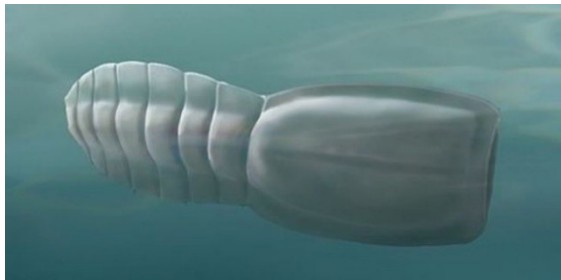

**Figure 9.** *Nesonektris aldridgei*. Image credit: Modified from D. García-Bellido. Used here per CC BY-SA 4.0.

The animal attained a length of 1.7 cm. An "axial, rod-like structure in the posterior body region," filling much of the "tail cavity," has been interpreted as a notochord [46], thus lending support to the deuterostome interpretation of vetulicolians. An alternate interpretation is that the thick rod structure represents a wide posterior gut region, although an alimentary tract interpretation would be difficult to reconcile with fragmentation of this linear structure into segments intepreted as notochordal discs [46].

### 4.8. Ooedigera

*Ooedigera's* relatively large size (up to 42 mm) and tuberculate ornamentation on its anterior section sets it apart from other vetulicolians. *Ooedigera* (Figure 10) is known from the lower Cambrian Sirius Passet Lagerstätte of Greenland [52]. If the tuberculate ornamentation is considered a juvenile trait, as in the possible juveniles *Beidazoon* and *Bullivetula*, then *Ooedigera* might be considered to be a neotenous form.

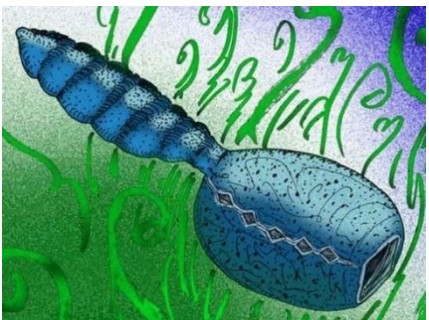

**Figure 10.** *Ooedigera*. Length of animal 4.2 cm. PaleoArt credit: Apokryltaros. Used here per CC BY-SA 3.0.

### 4.9. Pomatrum

*Pomatrum ventralis* has a typical overall vetulicolian body form, with a chicken-egg-shaped anterior portion and a flattened, beaver-tail-like posterior section or 'tail'. *Pomatrum* (Figure 11) is one of the 'soft shell' vetulicolians [49]. *Pomatrum* is remarkable for its circlet mouth consisting of concentric zones of 30 or more plates. Five gill structures ($G_1$–$G_5$) are present on either side of the anterior portion [49]. The anterior section is smooth apart from some transverse grooves. The posterior region has ten or more segments, with the anterior-most partitions becoming fainter and more closely spaced [49].

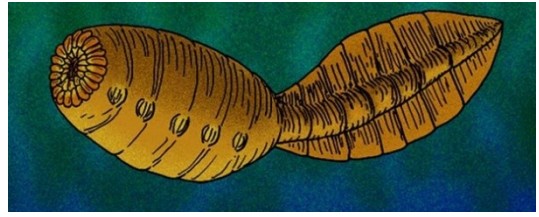

**Figure 11.** *Pomatrum*. Stanton F. Fink. Used here per CC BY-SA 2.5.

### 4.10. Skeemella

*Skeemella*, one of the strangest animals assigned to the vetulicolians, occurs in Middle Cambrian strata of Utah, USA [53]. *Skeemella* has a relatively small anterior section and a highly elongated posterior section that is divided into dorsal and ventral plates or half-rings. A telson-like, double-pointed or forked posterior tip occurs at the end of the vermiform posterior section. This structure is presumed here to be homologous to the caudal notch of *Banffia*.

*Skeemella's* placement in the Vetulicolia is controversial. *Skeemella* has a general morphology that is similar to the 7–8 stage Sytox green fluorescence nuclear-stained embryo of the millipede *Strigamia* [54], and the resemblence may not be entirely superficial if both *Skeemella* and *Strigamia* had dorsal expression of the segment polarity gene *engrailed*. *Skeemella* is nevertheless considered here

to be both deuterostome and vetulicolian, with the dorsal-ventral partition in the posterior region, with 'lateral line' partition between them on both sides of the animal, representing an extension of the 'gill'-bearing lateral line structure seen on the anterior end of vetulicolians. Some type of *Hox*-related genetic modification has evidently led to caudal extension of the vetulicolian lateral line in *Skeemella*.

### 4.11. Vetulicola

The type species of the genus is *Vetulicola cuneata* [50], and it remains the best known species in the genus. *Vetulicola* (Figure 12) is one of the 'hard shell' vetulicolians [49]. *Vetulicola* has a 'quadrate carapace' [55] with a ventral keel, a pointed dorsal fin, and a segmented, dorsally advanced arthropod-like posterior section. Spindle-shaped gill structures are present, showing what are interpreted as gill pouches, gill filaments, and gill slits [4,15,49]. Up to three orders of faint annulations occur on the anterior body section. Seven segments are seen on the posterior section. *Vetulicola rectangulata* has a relatively slender posterior region (Figure 12) but it is otherwise similar to *V. cuneata*. *Vetulicola longbaoshanensis* from the Gushan Biota has more elaborate gill structures [15], somewhat resembling those of an undescribed new taxon of vetulicolian from the Qingjiang Biota ([16], Supplementary Materials).

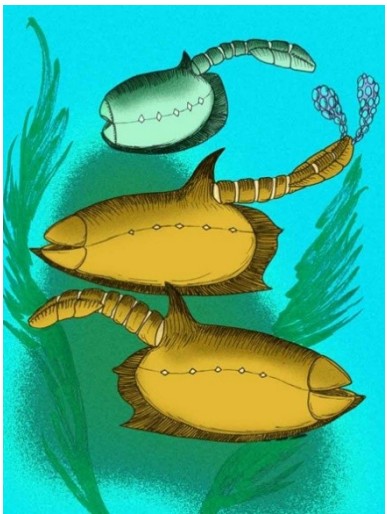

**Figure 12.** *Vetulicola rectangulata* (top specimen) and *Vetulicola cuneata* (center and bottom specimens). Note, the wedge beak in *Vetulicola cuneata*. PaleoArt credit: Apokryltaros. Used here per CC BY-SA 3.0.

### 4.12. Vetulicolian gen. et sp. Indet. A

This vetulicolian, 28 mm in length, has a posterior section that is similar to *Ooedigera peeli* but "with a broader flattened area ventrally than dorsally" [52]. The single known specimen has a relatively smooth surface.

### 4.13. Xidazoon

*Xidazoon* [56] is a possible junior synonym of *Pomatrum*.

### 4.14. Yuyuanozoon

Known only from the holotype, *Yuyuanozoon magnificissimi* (Figure 13) is another vetulicolian from Chengjiang [43]. Its anterior section is ovoid and comparatively smooth, with five pairs of gill structures arranged in a chain-like array on either side of the anterior section of the animal. *Yuyuanozoon* reaches over 20 cm in length. As typical for vetulicolians, its posterior section is divided into seven segments. The posterior section is cylindrical, rather than being flattened as in some other vetulicolians. The anterior section has six segments.

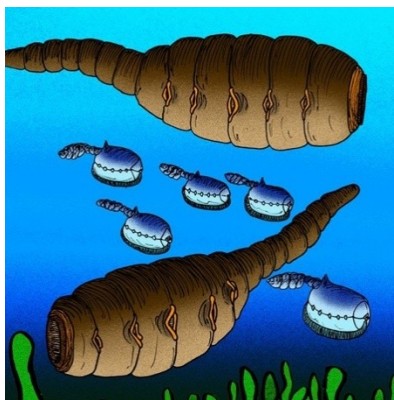

**Figure 13.** *Yuyuanozoon magnificissimi* with smaller *Vetulicola rectangulata* swimming behind. PaleoArt credit: Apokryltaros. Used here per CC BY-SA 3.0.

The large mouth region [57,58] suggests that *Yuyuanozoon* may have been well suited to filter feeding strategies similar to those of a modern salp or doliodid, although any similarities are likely to be superficial. An unidentified modern doliolid (possibly genus *Doliolum*) bears resemblance to the anterior section of *Yuyuanozoon magnificissimi* [Figure 13], which might imply a filter feeding lifestyle for the latter.

## 5. *Shenzianyuloma* nov. gen.

A new species of vetulicolian (*Shenzianyuloma yunnanense* nov. gen. nov. sp.) from the early Cambrian Chengjiang Biota (Burgess Shale-Type (BST) deposit; Maotianshan Shale, 518 Ma), represents the earliest example of 'angelfish' body form in a nectobenthic deuterostome (Figures 14–24). Although the exact locality of the holotype is unknown, a specimen of the brachiopod *Diadonga pista* occurs on the same slab (Figure 14), in accordance with matrix characteristics (yellow shale), indicating that the specimen is indeed derived from the Maotianshan Shale.

The fossil was acquired in February 2019 from a crystal and fossil vendor located in Lianyungang, Jiangsu, China. This online purchase rescued an important specimen that was at risk of being lost to science by vanishing into a private collection. The specimen (IGM 5008) is now available for study by the paleontological research community.

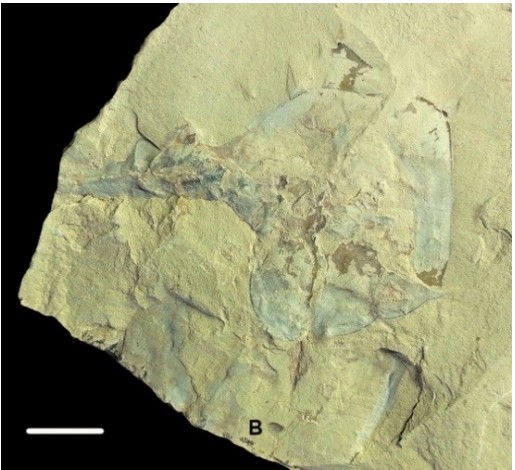

**Figure 14.** *Shenzianyuloma yunnanense* nov. gen. nov. sp. Holotype. Sample 1 of 6/25/2019; IGM 5008. **B**, Epifaunal brachiopod *Diadonga pista*. Scale bar 1 cm.

*Shenzianyuloma yunnanense* nov. gen. nov. sp. is preserved as a compression fossil in yellow Chengjiang shale. The perimeter/margins of the fossil seem to be intact, suggesting that the animal in life had a slender profile as viewed from the front or back. The proximal part of the tail and midpoint of the flanks experienced minor deformation that led to displacement of some of the gill structures and, significantly, de-nesting of one or more of the myomere cones. The gill structures have the shape of ovoid capsules, and one of these may have split apart, allowing gill filament-like structures to project downward (Figures 17 and 18).

In spite of the central disturbance, the fossilized animal is largely intact and gives a good representation of the shape of the creature. *Shenzianyuloma yunnanense* nov. gen. nov. sp. is not a decayed or mutilated specimen of a known taxon.

The rock sample bearing the fossil has dimensions 12 cm × 7.8 cm, and is 1.4 cm thick. The preserved length of the animal is 5 cm, with an anterior body region 4 cm deep (Figures 14 and 15). It has a dorsal fin (Figures 14 and 15) with a saw tooth margin. The anterior of the animal is blunt, with a ventral beak.

Although its posterior is not segmented, *Shenzianyuloma yunnanense* nov. gen. nov. sp. has four clearly defined segments in the ventral part of its anterior portion, whereas the dorsal part of the anterior section where the dorsal fin attaches is smooth, almost domal.

Segment 1 of the ventral anterior forms the 'mandible' of the beak. This does appear to be a beak-like structure, as opposed to being some sort of spinose projection, due to its similarity to the 'wedge beak' known from *Vetulicola cuneata*. Segments 2–4 increase in depth in a posterior direction, with segment 4 extending downward to form the edge of a ventral keel. The posterior section seems to continue as an extension of this segmented ventral part of the anterior section. This would be inverse to the pattern proposed for *Heteromorphus* subtype new species Form A [49], where the posterior section is thought to be likely an extension of the dorsal half of the anterior section.

The dorsal part of the anterior section appears to be unsegmented, rather developing a triangular fin that mirrors the profile of ventral segments 1–4 to give the angelfish body shape. The dorsal fin has blunt teeth on its leading edge that increase in size in an anterior direction. The base of the dorsal fin develops a thin dark line that parallels the fin insertion at the top of the animal's back (Figure 16). The line is approximately 0.4 mm from the base of the fin.

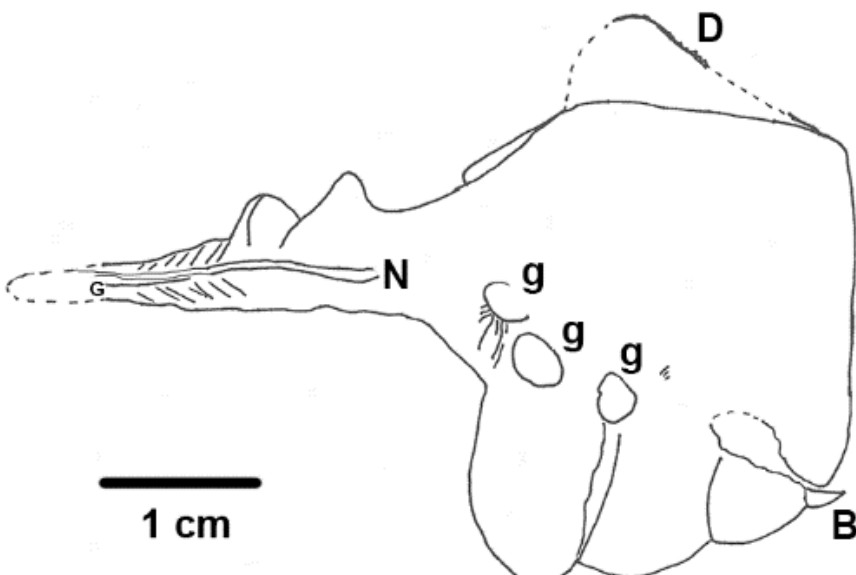

**Figure 15.** *Shenzianyuloma yunnanense* nov. gen. nov. sp. IGM 5008. Line art sketch showing morphology of the specimen. **B**, wedge beak; **D**, dorsal fin with denticulate leading margin; **g**, gill structure; **G**, presumed gut trace; **N**, notochord. Scale bar 1 cm. Image Credit: Mark McMenamin.

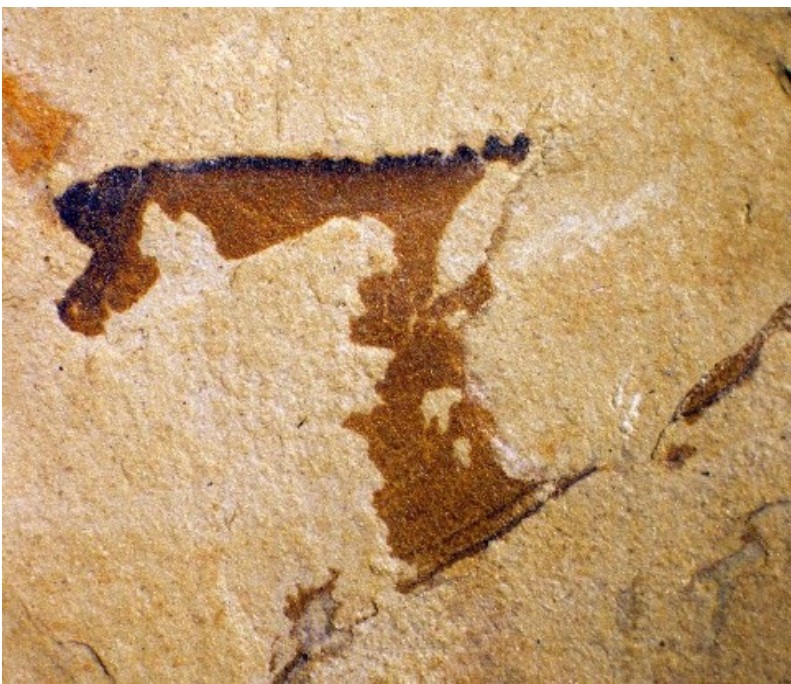

**Figure 16.** *Shenzianyuloma yunnanense* nov. gen. nov. sp. IGM 5008. Dorsal fin detail showing denticulate margin and thin ridge that parallels the base of the fin. Photo Credit: Mark McMenamin.

Three (of a presumed original five) gill structures are preserved at the interface between the segmented ventral and smooth dorsal parts of the anterior section (Figure 15). The posterior-most reserved gill structure preserves paired gill filaments that trail downward from the gill structure (Figures 17 and 18).

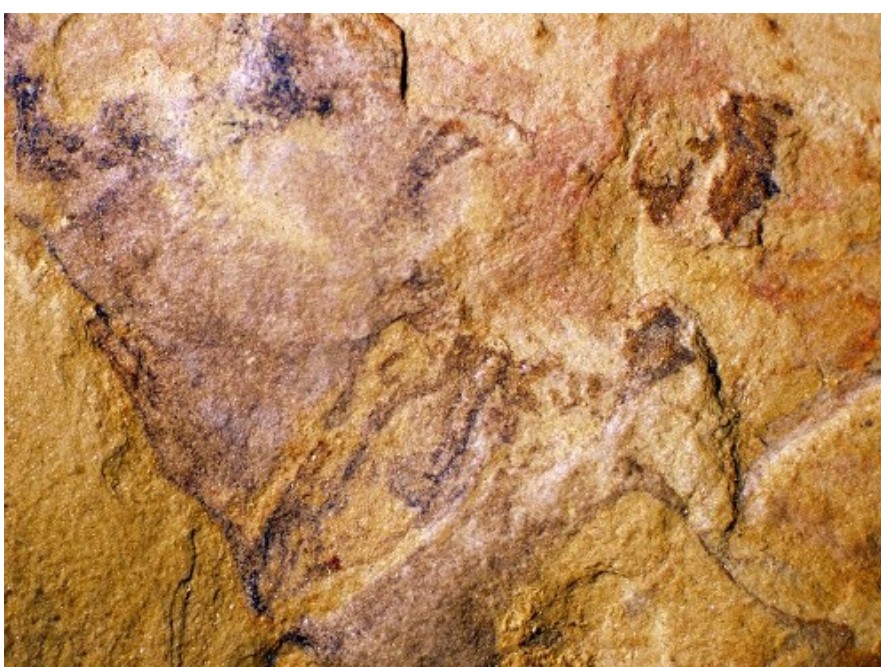

**Figure 17.** *Shenzianyuloma yunnanense* nov. gen. nov. sp. IGM 5008. Detail of filamentous gill structure. Photo Credit: Mark McMenamin.

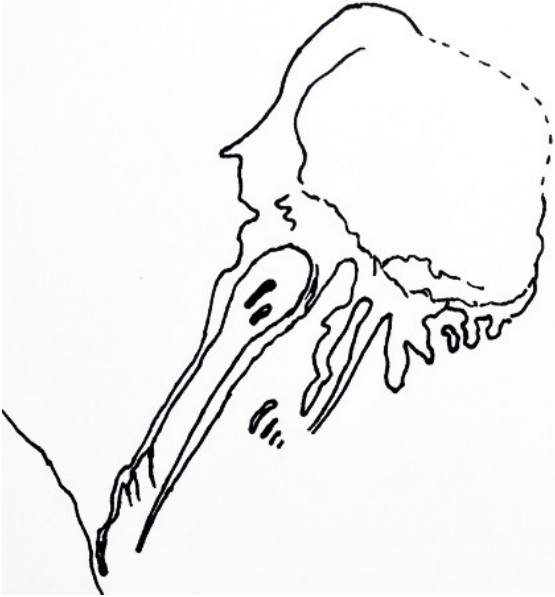

**Figure 18.** *Shenzianyuloma yunnanense* nov. gen. nov. sp. IGM 5008. Sketch of gill structure and filaments in previous image. Image Credit: Mark McMenamin.

The posterior section of *Shenzianyuloma yunnanense* nov. gen. nov. sp. provides key information about the affinities of this creature. The 'tail' shows evidence for both a notochord and a distinct posterior gut trace. Both the notochord and the gut trace are surrounded above and below by anteriorly slanting myotomes (Figures 19–22).

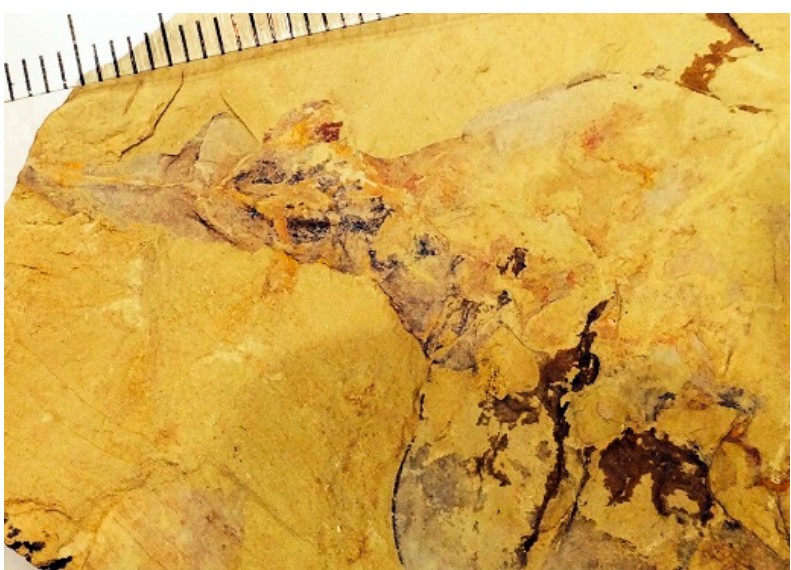

**Figure 19.** *Shenzianyuloma yunnanense* nov. gen. nov. sp. IGM 5008. Posterior region of animal. Scale in mm. Photo Credit: Mark McMenamin.

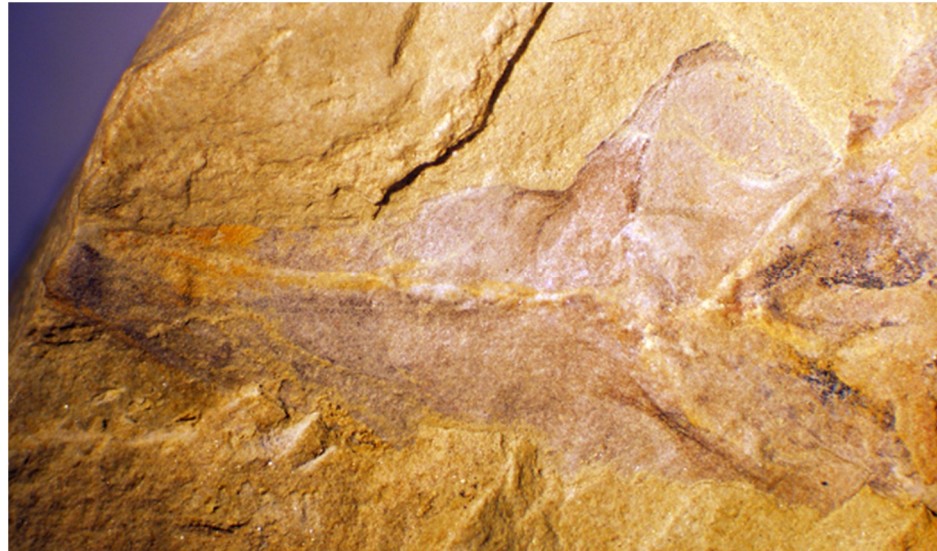

**Figure 20.** *Shenzianyuloma yunnanense* nov. gen. nov. sp. IGM 5008. Detail of tail region showing notochord and posterior gut trace. Photo Credit: Mark McMenamin.

The notochord is preserved as a light-colored band that runs along the dorsal half of the tail. Its width is approximately 3–4 mm diameter, and about 1.3 cm of notochord length is preserved (Figures 19–22). Two prominent nodes along its length (to the right of center in Figure 22) likely represent notochordal discs [46].

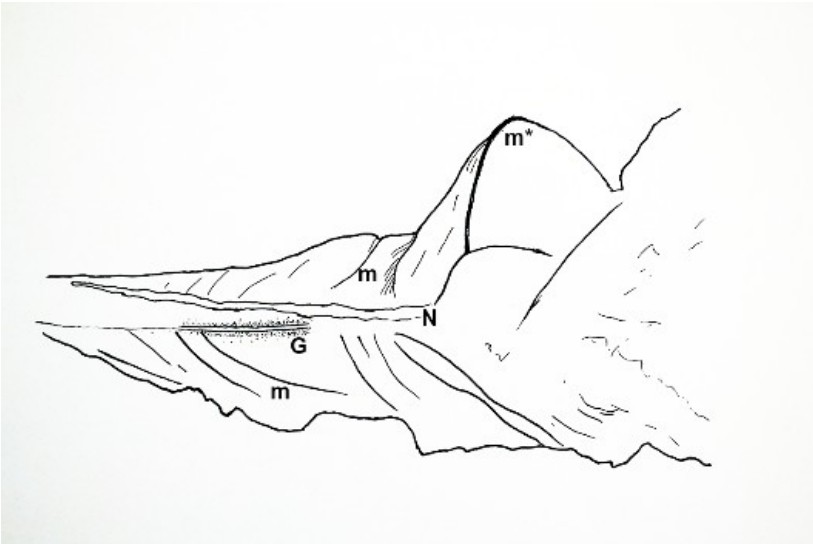

**Figure 21.** *Shenzianyuloma yunnanense* nov. gen. nov. sp. IGM 5008. Sketch of tail region from previous image. **G**, presumed gut trace; **m**, myomeres; **m***, delaminated myomere; **N**, notochord. Image Credit: Mark McMenamin.

The gut trace is preserved as a linear structure with 7 mm preserved length. It is approximately 0.35 mm in width with a distinct, very narrow central canal about 0.05 mm wide. Fine linear structures project at high angles to the central tube of the alimentary tract (Figures 22 and 23); these are interpreted here as gut diverticula [47]. They are definitely projections from the central gut trace, rather than being cracks or fractures of some type in the vicinity of the gut.

The diverticula connect to the main gut trace at a variety of high angles (Figure 23). In *Banffia*, diverticula usually connect with the intestinal trace at approximate right angles, but not infrequently, are offset at high angles [47] similar to the arrangement seen in Figure 23.

A similar morphology is seen in *Pomatrum* cf. *ventralis*, specimen YKLP 10914a [50], where an alimentary tract is ventral to a linear structure flanked by "numerous transverse lines" [50] that may very well represent a notochord surrounded by myotome cones.

Disturbance to the central part of the animal displaced gill structures and caused delamination or de-nesting of the myotome cones in the tail region (**m\*** in Figure 21). This was a key occurrence, as it demonstrated the presence of cone-in-cone myomere structure in *Shenzianyuloma yunnanense* nov. gen. nov. sp. Connective tissues were apparently not yet as strong as those in a modern fish, allowing the delamination to occur.

Muscle fibers are visible in the myotome immediately posterior to the delaminated myotome, just to the right of the middle **m** in Figure 21.

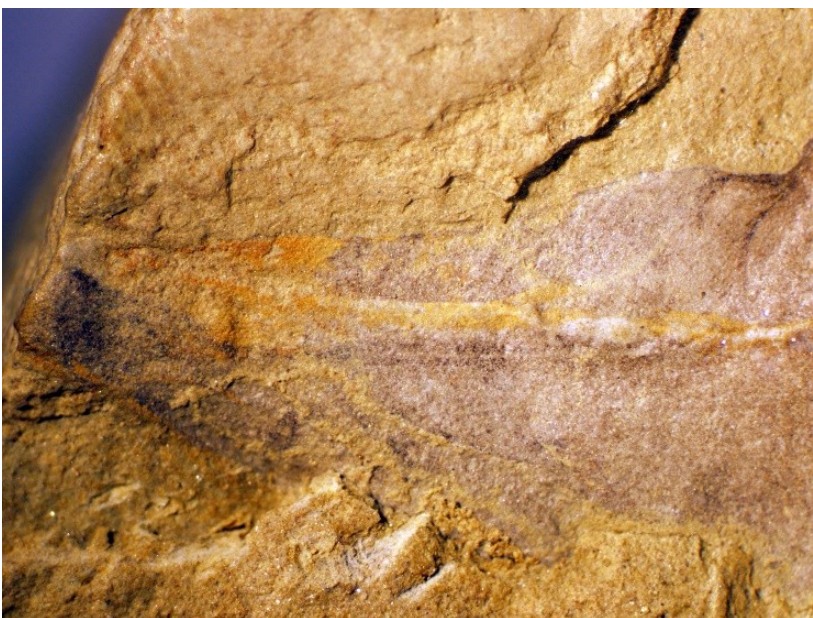

**Figure 22.** *Shenzianyuloma yunnanense* nov. gen. nov. sp. IGM 5008. Detail showing notochord and posterior gut trace. Myomeres are visible in both the dorsal and ventral portions of the tail. Photo Credit: Mark McMenamin.

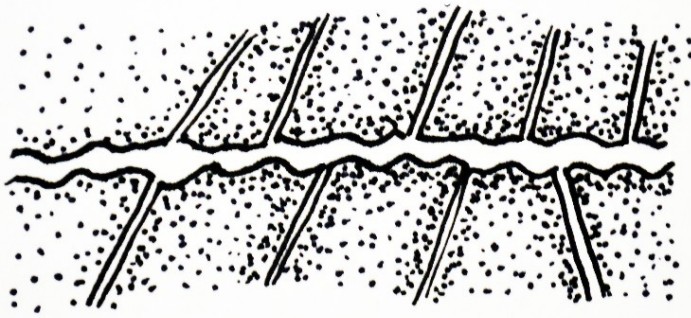

**Figure 23.** *Shenzianyuloma yunnanense* nov. gen. nov. sp. IGM 5008. Detail of alimentary tract and gut diverticula at the anterior end of preserved gut trace. Width of view 700 microns. Image Credit: Mark McMenamin.

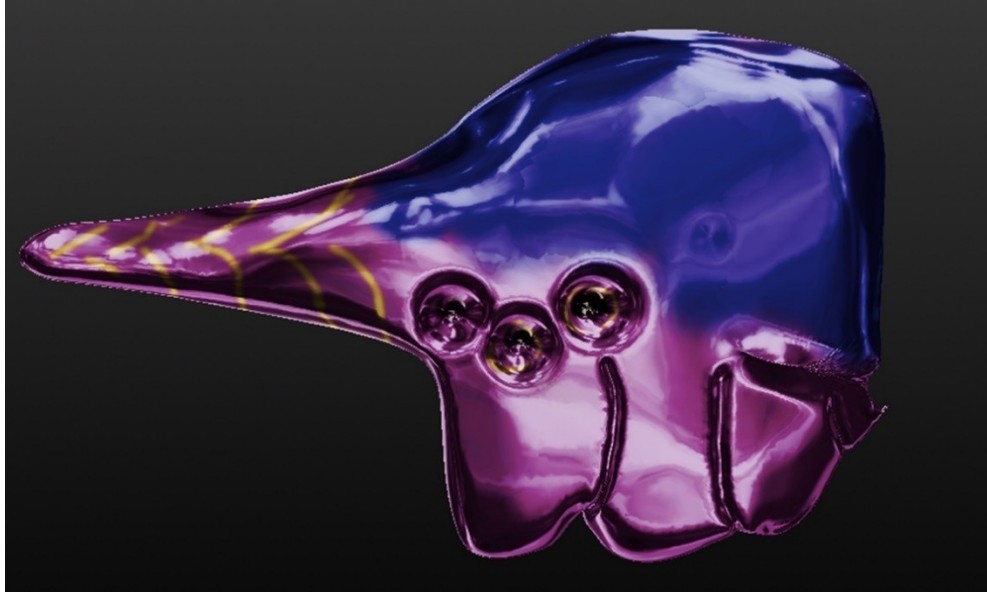

**Figure 24.** *Shenzianyuloma yunnanense* nov. gen. nov. sp., digital reconstruction. This partial reconstruction omits the dorsal fin and any conjectural tail/caudal 'fin,' as the posterior tip of the animal is unknown. Image Credit: Mark McMenamin, using Sculptris Alpha 6 by Pixologic, Inc.

*Vetulicolian Locomotion*

The seemingly awkward body shape of virtually all vetulicolians begs the question of how these creatures were able to swim and to maintain their position in the water column. The roughly triangular, angelfish-like body form of angelfish vetulicolian (*Shenzianyuloma yunnanense* nov. gen. nov. sp.), complete with a protruding lower 'jaw,' is well known from the freshwater angelfish *Pterophyllum*, the marine angelfish *Pomacanthus*, and a variety of fossil fishes including *Eoplatex papilio* and the pycnodont *Proscinetes*. Lacking pectoral fins or their functional equivalents, *Shenzianyuloma yunnanense* nov. gen. nov. sp. would presumably have been incapable of labriform swimming, and would have relied instead on less-efficient tail propulsion. Jets of water from the gill structures may have also helped to propel angelfish vetulicolian forward. The dorsal fin on the anterior section, as well as its denticulate leading margin, imply at least a certain degree of controlled forward motion and hydrodynamic stability in the water column.

## 6. Developmental Biology of Chordate/Vetulicolian Origins

Several aspects should be mentioned with regard to developmental biology when considering the origin of Chordata and related Cambrian animals. For example, the distinction between the tiny pharyngeal teeth of *Haikouella* and the larger pharyngeal teeth of *Yunnanozoon*, homologous features [7] between the two genera, represents the range of variation seen in the early chordate anterior scleritome.

The body division into two halves in vetulicolians has been invoked as an illustration of Romer's somato-visceral theory [51,59]. It seems reasonable to speculate that a discontinuity in all-*trans*-retinoic acid (ATRA)-FGF morphogen signaling molecules has occurred at the vetulicolian midline; thus, in vetulicolians, the anterior part of the animal and the posterior part of the animal may have been dominated by different molecular signaling regimes. For example, a dual retinoic acid gradient is involved in down-regulating FGF8 to permit somitogenesis. Interestingly, the gradient is at a minimum value at the head/tail junction, and reaches a maximum in the anterior section [60].

The (ATRA)-FGF double-sided retinoic acid gradient signal isolation seems to be particularly pronounced in *Skeemella* [53]. Somite/segmentation formation is dramatically favored in the posterior section and more subdued (although still present) in the anterior section, suggesting that the same posterior-generated signaling factor is influencing segmentation in both the posterior (influence strong)

and anterior (influence weak) sections, with the presumed ATRA gradient dropping off sharply across the anterior–posterior sections division [61]. Here, we see a potential unification of the morphogenetic field and gradient signaling approaches, with a morphogenetic field discontinuity demarcating fields of morphogen signaling factor influence.

The serial gene duplications known to have occurred in vertebrates, gnathostomes and teleost fish evidently cannot be linked to the "evolution of complexity in vertebrates" [29].

## 7. Conclusions

Among living chordates, molecular phylogeny shows that tunicates are more closely related to vertebrates (they jointly form the Olfactores) than vertebrates are to lancelets (Cephalochordata [61,62]). Curiously, in the latter analysis, "the monophyly of deuterostomes remained poorly supported" [62]. The reliability of molecular clocks based on these data ("most major lineages of deuterostomes arose prior to the Cambrian Explosion" [61]) is open to question, considering that few if any deuterostome fossils are known from strata deposited before the Cambrian boundary. In any case, the general consensus is that vertebrates first appeared in the Cambrian [63].

A putative exception to this general rule of Cambrian chordate appearance is the identification of *Burykhia hunti* from Ediacaran strata of the White Sea region, Russia and *Ausia fenestrata* from the Nama Group in Namibia, as the oldest known ascidians [64]. It seems highly unlikely that members of the Ausiidae are urochordates, considering that their morphology consists of tubes or bags with serial perforations. The presence of a zig-zag medial suture in *Burykhia* suggests rather that the fossils are more properly allied with frondose 'vendobiont' Ediacarans such as *Charnia*, *Rangea*, *Phyllozoon* and *Pteridinium*.

It seems quite plausible that the advent of deuterostomes near the Cambrian boundary involved *both* a reversal of gut polarity and a two-sided retinoic acid gradient, with a gradient discontinuity at the midpoint of the organism that is reflected in the characteristic division of vetulicolians into morphologically distinct anterior and posterior sections.

The posterior section of *Shenzianyuloma yunnanense* nov. gen. nov. sp. demonstrates that vetulicolians are indeed deuterostomes allied to chordates [65], due to *Shenzianyuloma's* notochord (with discs) in a position that is dorsal to an alimentary tract with diverticula, and also due to the presence of myotome cones [66] with preserved muscle fibers.

It is now possible to review the current state of knowledge regarding Cambrian chordates, vetulicolians and their affinities. All of these animals are deuterostomes (i.e., the blastopore becomes the anus during early ontogeny), although of course, other deuterostomes are also known from the early Cambrian (cambroernids, echinoderms and hemichordates).

Three groups may be recognized among the Cambrian chordates. Urochordates, characterized by the presence of a branchial basket, are represented by *Cheungkongella* and *Shankouclava*. Cephalochordates, characterized by a full notochord, are represented by four genera: *Cathaymyrus Pikaia* (with question), *Zhongjianichthys*, and *Zhongxiniscus*. Vertebrates, characterized by a neural crest, are also represented by four genera: *Haikouella*, *Haikouichthys*, *Metaspriggina* and *Myllokunmingia*.

Vetulicolians are characterized by their bipartite body form and by five pairs of gill structures. Two informal groups of vetulicolians can be recognized, the "soft shell vetulicolians" and the "hard shell vetulicolians". Soft shell vetulicolians lack segmentation in their posterior section or tail; lack a prominent constriction between the anterior and posterior portions of the body; and have a very blunt, rectangular anterior. Hard shell vetulicolians have a segmented posterior section, and may have a tuberculate anterior section, at least as juveniles.

Soft shell vetulicolians are represented by: *Didazoon*, *Heteromorphus*, *Pomatrum* and *Xidazoon*. Hard shell vetulicolians are represented by six genera: *Beidazoon*, *Bullivetula*, *Nesonektris*, *Ooedigera*, *Vetulicola* and *Yuyuanozoon*.

*Skeemella* and *Shenzianyuloma* are unique, unusual vetulicolians that cannot be placed in either the soft shell or hard shell clade. *Yunnanozoan*, although surely a bilaterian deuterostome, cannot be confidently placed in either the chordates or the vetulicolians.

Taken together, these chordate and vetulicolian genera document an important part of the explosive radiation of deuterostomes at the base of the Cambrian. Although debate is likely to continue, the preponderance of the evidence does suggest that vetulicolians and yunnanozoans are indeed deuterostomes [47,50,52,53,67–71]. Although deuterostomes, the extremely odd character set in vetulicolians also suggests that they may very well represent a higher taxon of phylum rank. Great caution, however, is needed in order to avoid circular reasoning with regard to the use of gross morphology to make such phylogenetic claims. The five pairs of gill structures in vetulicolians could in fact represent mid-gut glands of some sort, although this seems unlikely. The bizarre carpoids, with (like vetulicolians) a bipartite body division are now considered to be regular echinoderms with no gill slits and a feeding arm with a normal water vascular system [72]. Still, the mythical chimaera-like nature of many of the early deuterostomes suggests that the group may very well require the erection of multiple extinct phylum-level taxa to avoid systematic shoehorning.

Aldridge et al. [50] note that although the vetulicolians "appear to form a monophyletic clade, it is premature to accord them phylum rank without resolution of their phylogenetic position." This remains a cogent observation, for indeed, with *Shenzianyuloma yunnanense* nov. gen. nov. sp. displaying evidence for conical myomeres surrounding a notochord with discs, the possibility that *Shenzianyuloma* and other vetulicolians represent stem chordates must be seriously considered.

## 8. Systematic Paleontology

Repository data: IGM, Institute of Geology Museum, Departmento de Paleontología, Cuidad Universitaria, Delegacíon de Coyoacán, 04510, México; YKLP, Key Laboratory for Palaeobiology, Yunnan University, Kunming, China.

Kingdom Animalia
Superphylum Deuterostomia
Phylum Vetulicolia

**Genus *Shenzianyuloma* nov. gen**.

**Diagnosis:** Vetulicolian with a body divided into a broad, laterally flattened anterior section and relatively narrow posterior or tail region. The anterior section is divided into a relatively smooth dorsal part with a denticulate dorsal fin and a segmented or partitioned ventral part. The dorsal fin has a thin ridge that parallels the base of the fin. A wedge beak occurs at the anterior junction of the dorsal and ventral parts. Five gill structures with filaments occur between the dorsal and ventral parts of the anterior section. The dorsal and ventral parts of the anterior section have approximately similar (but inverted relative to one another) profiles. The unsegmented posterior or tail region develops a notochord, a gut trace with diverticula, and at least seven myomeres. The posterior tip of the tail region is unknown.

**Description**: This vetulicolian has an unsegmented posterior section ('tail') and anterior section that is segmented only in its ventral part, where four segments occur. Gill structures occur between the dorsal and ventral parts of the anterior section. At the anterior of the animal, a wedge beak forms from segment 1 at the juncture between the dorsal and ventral anterior section parts. The sagittal crest bears a dorsal fin with a denticulate anterior margin. The dorsal fin has blunt teeth on its leading edge that increase in size toward the front of the fin. The base of the dorsal fin shows a thin dark linear ridge that parallels the fin insertion at the top of the animal's back, at a distance of 0.4 mm from the fin base (Figure 16).

A thin keel forms the ventral margin of the anterior section, and the anterior-ventral profile roughly matches that of the main dorsal fin. The dorsal part of the narrow, dorsally advanced posterior section bears three fins, two larger and flap-shaped and a posterior one that is much smaller.

The posterior section has both a notochord (with notochordal discs) and distinct gut trace (with diverticula). Both notochord and gut/intestine are surrounded by conical myotomes. Extremely fine lines project at approximate right angles to the central tube of the alimentary tract (Figure 22); these represent gut diverticula [44].

**Discussion**: The posterior tip of *Shenzianyuloma yunnanense* nov. gen. nov. sp. is unknown due to breakage in the holotype. The presence of myotomes strongly suggest that *Shenzianyuloma yunnanense* nov. gen. nov. sp. is both a deuterostome and a chordate.

*Shenzianyuloma yunnanense* nov. gen. nov. sp. resembles "*Heteromorphus* subtype new species Form A" [46] in being a 'soft shell' vetulicolian form that: lacks segmentation in its posterior section or tail; lacks a prominent constriction between the anterior and posterior portions of the body; and has a very blunt, rectangular anterior. *Shenzianyuloma yunnanense* nov. gen. nov. sp. differs from "*Heteromorphus* subtype new species Form A" by having: a narrower tail that lacks wrinkles; anteriorly slanting myotome-like structures; a wedge beak; and several flap- or scoop-like fins along the dorsal edge of its posterior section. A gut trace extends to the preserved end of the tail in *Shenzianyuloma yunnanense* nov. gen. nov. sp. As in *Haikouella* and *Myllokunmingia*, gill filaments that proceed from the gill structure in *Shenzianyuloma* appear to be paired.

Sagittal plane expansion and lateral flattening of the anterior section of *Shenzianyuloma* provides a striking morphological contrast to the dorsal-ventral flattening of the anterior section of the Burgess Shale vetulicolid *Banffia constricta* (Figure 7), thereby expanding our knowledge of the range of form in this important group of early deuterostomes.

The dorsal fin in *Shenzianyuloma* is comparable to the dorsal fin of *Myllokunmingia*, although *Myllokunmingia's* dorsal fin has a comparatively lower profile [4] and lacks the denticles on the leading edge of the fin.

*Shenzianyuloma yunnanense* nov. gen. nov. sp. resembles "*Heteromorphus* subtype new species Form A" [46] in being a 'soft shell' form that: lacks segmentation in its posterior section or tail; lacks a prominent constriction between the anterior and posterior portions of the body; and has a very blunt, rectangular anterior end. *Shenzianyuloma yunnanense* nov. gen. nov. sp. differs from "*Heteromorphus* subtype new species Form A" by having: a narrower tail that lacks wrinkles, but has anteriorly slanting myotome-like structures, and a wedge beak. A gut trace extends to the preserved end of the tail in *Shenzianyuloma yunnanense* nov. gen. nov. sp. As in *Haikouella* and *Myllokunmingia*, gill filaments that proceed from the gill structure in *Shenzianyuloma* appear to be paired.

A unique 'dorsal fin' in *Shenzianyuloma yunnanense* nov. gen. nov. sp. mirrors the profile of the ventral cuticular plate. The anterior end of the latter forms a 'wedge beak' similar to that described from *Vetulicola cuneata*. The wedge beak-bearing cuticular plate consists of three articulating sections, implying a dentary-like mobility in what appears to be the earliest beak-like 'jaw' structure reported from a deuterostome. Whether or not *Shenzianyuloma yunnanense* nov. gen. nov. sp. also has a vertical mouth ('M' in [46]) as inferred for "*Heteromorphus* subtype new species Form A" is unknown. If so, *Shenzianyuloma* may have had a dual oral apparatus, a biting beak at the lower part of the mouth (i.e., jaw with horizontal opening) and a blunt, elongate 'M' mouth opening that forms a vertical gape for the purposes of filter feeding.

The anteriorly slanting myotome-like structures of *Shenzianyuloma yunnanense* nov. gen. nov. sp. strongly support the hypothesis that vetulicolians are deuterostomes rather than arthropod-like protostomes.

**Etymology**: The genus name is a neuter noun formed as a combination and rearrangement of the Chinese word for angelfish (Shénxiān yú, 神仙鱼) and an anagram of the genus name for the ocean sunfish, *Mola*.

***Shenzianyuloma yunnanense* nov. gen. nov. sp.**: Figures 14–24

**Holotype**: Field sample 1 of 6/25/2019; IGM 5008.
**Description**: Same as for genus, by monotypy.

**Etymology:** The trivial name, in honor of Yunnan, China, is rendered here "*yunnanense*" rather than "*yunnanensis*" to agree in gender with the neuter noun *Shenzianyuloma*.

**Age and Locality Information:** Early Cambrian Chengjiang Biota (Burgess Shale-Type (BST) deposit; Maotianshan Shale, 518 Ma).

**Supplementary Materials:** The following are available online at http://www.mdpi.com/2076-3263/9/8/354/s1, Supplemental file: Statement on provenance of Shenzianyuloma holotype. Video S1: fossil specimen and 3D model reconstruction.

**Acknowledgments:** Thanks to Dianna L. Schulte McMenamin, Jon Tennant and three reviewers for assistance with this research.

**Conflicts of Interest:** The author declares no conflict of interest.

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
