# Peer review of "Cambrian Chordates and Vetulicolians"

_geosciences, doi:10.3390/geosciences9080354_

Round 1
Reviewer 1 Report
This paper is effectively formed of two sections. One is a (fairly) straightforward review of putative chordates and apparently related taxa from the Cambrian. The second is the description of what appears to be a new taxon of vetulicolian. For the most part the text is straightforward in terms of narrative, but has some excursions that are of questionable relevance to the overall theme of the paper. Finally there is a potential ethical issue, although this may well be of no consequence. In detail:
(a) The overview of the Cambrian chordates and vetulicolians (plus other taxa that may be deuterostomes but are of even more enigmatic status e.g. Yunnanozoon) is straightforward and as such provides a helpful catalogue. Unsurprisingly, it draws on a diverse literature and in some cases adopts a rather uncritical view, especially of various purported anatomical structures. A somewhat keener critique would help to ginger-up the paper.
(b) The description of the purported new vetulicolian, Shenxianyuloma yunnanense, is potentially of significant interest to the community. Unfortunately its real relevance is very difficult to judge from the available information:
(i) The illustrations do not do the material justice. In the single photograph the details are not easy to resolve (even when magnified) and at the least there need to be close-ups of the putative gills and myomeres.
(ii) The accompanying interpretative drawing is very impressionistic and needs to be "fleshed out" to achieve the clarity that is now routine in such illustrations.
(iii) The identification of myomeres is potentially of major significance given that the posterior section of other vetulicolians is quasi-arthropodan. As noted above it is, however, very difficult to see clearly these segmental structures. One might wonder, for example, if they are more similar to the multiple segments of Skeemella? I take it that there is no evidence for a notochord? Let us say, for the sake of argument, that the segments are in chevrons. First, why should this per se make them myomeral? Second, if they are myomeres (that is a cone-in-cone musculature) then how is the transition achieved from the quasi-arthropodal configuration of sclerites and "arthropodial" membranes?
(iv) The description of the putative gills is not easy to follow, not least because they are almost impossible to resolve in the figure. So far as I recall no other vetulicolian has "paired gill filaments that trail downward from the gill structure"; rather the filaments (or whatever they are) are enclosed by the "capsule" that defines the gill opening.
(v) The curious so-called "wedge beak" seems to be much more similar to a spinose projection (not dissimilar to other such structures seen in various vetulicolians) and the hypothesized function as some sort of jaw seems rather fanciful. So too it seems rather improbable that an animal would have the dual function feeding of a "jaw" and filter-feeding.
(c) At various points in the text the narrative goes off at tangents that do little to augment the paper. Most notable in this regard are those sections (especially lines 167-174, 271-272, 374-389) that invoke various sorts of genetic mechanisms in the context of observed anatomical structures. For all we know such mechanisms were in operation, but we have absolutely no evidence that they were and as such they are effectively circular arguments. If there was some sort of independent evidence in the fossils for such genetic mechanisms (as elegantly shown, for example, in Ortega-Hernandez and Brena's paper on trilobite tergites (PLoS ONE 7, e52623 [2012])) then inclusion of such a discussion would be reasonable, but as it stands it is purely speculative.
(d) There may be a question as to the provenance of this specimen, as it is not clear that it comes from any established research group in China? Was it collected by the author? What and where is the repository IGM?
More minor comments
(e) lines 45-51. It is hardly fair to blame Gould for getting Banffia back-to-front. First, at that time practically nothing was known about Banffia. Second, his purpose (such as it was) was an overview rather than a monograph? So too, I am not sure that Briggs & Fortey were especially concerned about the status of Banffia. So too it seems to be an exaggeration that these authors were in any sense "thwarted" by the recognition of the Vetulicolia.
(f) lines 175-176. This comparison to the coelacanths seems to be fanciful.
(g) line 183. "awkward"; according to whom?
(h) line 187. As I recall the current consensus places Canadaspis as quite basal, and far removed from the ostracods?
(i) lines 248 and 250. "tuberculate" better than "tubercled"?
(j) line 254. "egg-shaped"? Chicken, snake, snail ...?
(k) line 263. "curiously"; why?
(l) lines 266-268. To give Wikipedia this sort of credence in a main-stream scientific paper is not appropriate.
(m) line 276. "mango", "orca-like"?
(n) line 279. If tubercles are absent then this needs no mention.
(o) lines 298-301. The similarities between a salp/doliodid and Yuyannozoon would seem to be entirely superficial.
(p) line 302 (and elsewhere). To compare this ?vetulicolian to an angelfish seems a bit of a stretch.
(q) lines 314 ff. Much of the following section is difficult to follow. If the author wants to make detailed comparisons to Heteromorphus, then comparative illustrative material would help. In part this section will be helped by a much clearer (and labelled) interpretative drawing of the unique specimen.
(r) lines 354-363. Whilst the mode(s) of locomotion of vetulicolians remain open to debate, these comparisons to various fish seem to be fanciful.
(s) lines 370-372. Comparisons between Metasprigginia and Hupesuchus are entirely fanciful.
(t) lines 401-407. As these two Ediacaran taxa are so poorly understood but unlikely to be any sort of deuterostome there seems little point in mentioning them.
(u) line 448. Does an "informal name" form any part of a proper taxonomic description?
(v) lines 499-503 and Figure 16. It is not really clear what the basis of this phylogeny is; it seems to be highly impressionistic.
Author Response
Reply Reviewer #1
a. Additional critical comments have been added to the review.
b. i. Additional material added to the revision to better characterize the new taxon.
b. ii. New illustrations have been added of the gill structures, myomeres and alimentary tract.
b. iii. I now discuss the discovery of a notochord in the new specimen; see revised text. No, this fossil does not closely resemble Skeemella.
b. iv. The gill capsule may have been disrupted by the deformation of the central part of the animal, allowing the gill filaments to project downward (see revised text).
b. v. The anterior end of the ventral anterior part is indeed a ‘wedge beak’ rather than a spinose projection; see revised text.
c. Analysis of genetic mechanism(s) is now essential for developmental biology, and may be applied by inference to the paleontological sciences as well. These are not circular arguments, see:
McMenamin, M.A.S.; McMenamin, S.K. Homeotic genes, the antennapedia complex in the trilobite genome, and iterative evolution in nevadiid and bristoliid trilobites. In McMenamin, M. A. S. (ed.), 2001, Paleontology Sonora: Lipalian and Cambrian. Meanma
Press, South Hadley, Massachusetts, USA. ISBN 1-893882-14-4
which should have been cited by:
Ortega-Hernández J, Brena C (2012) Ancestral Patterning of Tergite Formation in a Centipede Suggests Derived Mode of Trunk Segmentation in Trilobites. PLoS ONE 7(12): e52623. https://doi.org/10.1371/journal.pone.0052623
d. This fossil was obtained from a rock and crystal dealer. The author recognized the scientific significance of the specimen immediately, and took steps to acquire the fossil before it vanished into a private collection, thereby becoming lost to scrutiny by the scientific community. The specimen has been donated to, and is curated by, the Institute of Geology Museum in Mexico City where it may be accessed by researchers throughout the world.
e. Language revised/moderated here.
f. reference to coelacanths removed
g. ‘awkward’ changed to ‘enigmatic’
h. reference to Canadaspis removed
i. ‘tubercled’ changed to ‘tuberculate’ throughout
j. changed to ‘chicken egg-shaped’
k. ‘curiously’ removed
l. Wiki quote removed
m. mango, orca, removed.
n. tubercles mention removed
o. salp/doliolid—text revised here
p. The angelfish comparison is quite apt, and helps to distinguish this form from other Cambrian animals.
q. additional text and images provided in revision
r. Comparisons to fish fanciful? Not so, indeed, modern and ancient fish provide a good starting point for analysis of vetulicolian locomotion.
s. mention of Hupesuchus removed
t. Agreed, these Ediacaran taxa are not chordates nor even deuterstomes, but they should be mentioned as a foil to the authentic appearance of authentic deuterostomes.
u. ‘informal name’ section removed
v. figure removed; affinities discussion moved to revised text
Reviewer 2 Report
This MS offers a very interesting and well-documented review of a series of Cambrian deuterostomes, providing relatively exhaustive and up-to-date listings and brief descriptions of all main chordate and vetulicolian taxa described from this time interval, as well as the description of a new, weird, angelfish-shaped(?) vetulicolian from the famous Chengjiang Lagerstätte, South China. There is no doubt that such a review will constitute a useful contribution for scholars and scientists interested in the Cambrian Explosion and the early diversification of deuterostomes.
However, this MS might constitute an even more helpful and thorough review, if several points were modified/corrected. I have noted three main possible lines of improvement: the first one can be easily tackled and concerns a few typos; the second one concerns the content of the MS itself and the way it is written; the third one is about references.
(1) Typos
- line 49 (and elsewhere in the text): please modify ‘Early Cambrian’ into ‘early Cambrian’, as the Cambrian System is now subdivided into four (international) series. Its ‘traditional’ tripartite subdividision into ‘early’, ‘middle’ and ‘upper’ Cambrian is now only informal (so, no capitals);
- line 56: word missing (« the »?) between « was » and « discovery »;
- line 89: please put « Eoredlichia » in italics;
- line 183: please modify « the eponymous genus Vetulicolia [sic]» into « the eponymous genus Vetulicola »;
- line 369: please modify « sleritome » into « scleritome »;
- line 422: please add « a » between « and » and « lower »;
- line 429: please modify « The presence (…) suggest » into « The presence (…) suggests »;
- line 569: please modify « GeoBios » into « Geobios »;
- line 591: please delete « 2006 » after « Caron, J.-B. »;
- line 610: please modify « Acta Geological Sinica » into « Acta Geologica Sinica »;
(2) Content (text and figures)
A first major problem with this review is that its main objective(s) is/are nowhere clearly stated, and the reader does not understand why this contribution focuses on these two particular lineages. In the abstract (lines 11-12), the statement that « Two Cambrian deuterostome groups, the chordates and the vetulicolians, are of particular interest for understanding the evolutionary dynamics of the Cambrian evolutionary event » should be rephrased. There are several problems with this sentence and, more generally, in the way the MS is constructed.
First, although the MS is supposed to provide a relatively exhaustive state of the art of the current knowledge on early chordates and vetulicolians, there is no discussion of alternative interpretations of vetulicolians. Possible affinities/similarities with arthropods are simply mentioned briefly on lines 186-190 (« The overall shape of vetulicolians is somewhat reminiscent of bivalve early crustacean Canadaspis from the Burgess Shale, a probable ancestor of the ostracods, known for their hinged carapaces that serve as bivalve shells. But the ressemblance to the anterior tagmosis of crustaceans forming their carapace is superficial. The lateral line in vetulicolians represents a potential plane of flexure and this may be a characteristic of their unusual body plan »), shortly on lines 199-201 (« Originally assigned to the annelids, a lack of appendages has proved to be an unsurmountable obstacle for assigning Banffia to Arthropoda »), and finally on lines 350-352 (« If the anteriorly slanting myotome-like structures of Shenxianyuloma yunnanense nov. gen. nov. sp. are indeed myotomes, this strongly supports the hypothesis that vetulicolians are deuterostomes rather than arthropod-like protostomes »).
Even if the author considers vetulicolians as definitive deuterostomes, for the sake of science (and, in particular, in the case of a review paper), the ongoing debate on their taxonomic affinities should be summarized somewhere in the MS. This review offers the reader a single interpretation (and it is only, through one or two sentences, that, in passing, we can grasp/guess that other interpretations have been probably also proposed). Deuterostome affinities of vetulicolians have been questioned in a series of recent papers. The pro and contra of these alternative interpretations should be at least discussed briefly in this review.
Overall similarity of vetulicolians with a tadpole or Romer’s somato-visceral organism could be purely superficial (as superficial as e.g., similarities between vetulicolians and bivalved arthropods) and do not necessarily have any phylogenetic implication(s). For example, the literature on Cambrian chordates and vetulicolians is crowded with references to ‘carpoids’ (or ‘calcichordates’), generally presented as basal-most echinoderms with putative gill slits and a bipartite organization (with a head and a tail) similar to that of vetulicolians. Recently described stylophorans with their exceptionally preserved soft parts definitively showed that these fossils were ‘normal’ echinoderms with a feeding arm (with a regular water-vascular system) and no gill slits (Lefebvre et al. 2019, Geobios), and that their bipartite organization was only superficially similar to that of a tadpole or a vetulicolian (no head, no tail, but a theca and a feeding arm). This example is only to point out that comparisons based on superficial outlines / gross morphology should be considered with the most extreme care.
Papers by Butterfield (2003), Briggs et al. (2005) [cited as publication n°50], Caron (2006) [44], Aldridge et al. (2007) [47] and, to a certain extent, Vinther et al. (2017) [49] all question the phylogenetic position of vetulicolians within or outside of deuterostomes. The same cautious and scientific approach was followed by Cong et al. (2015) in their paper on the palaeobiology of yunnanozoans (a nice paper, not even cited in this MS, although Yunnanozoon is treated on lines 147-162). At least some of the issues raised in these papers should be expressed (and discussed) in the MS: e.g. alternative interpretations of the series of three-dimensional structures as gill slits vs. mid-gut glands, or presence of a cuticle (shell/carapace) in vetulicolians [a carapace is not really a deuterostome character], … Most (all) structures supporting the assignment of vetulicolians to deuterostomes (e.g., endostyle, gill slits, myotomes, …) entirely depend on the interpretation of these fossils as deuterostomes (see e.g. lines 350-352): we are close to circular reasoning and/or systematic shoe-horning.
Even if –as suggested by many authors- vetulicolians were actually deuterostomes, there is currently no agreement at all in the literature about their phylogenetic position within the deuterostomes (stem deuterostomes, basal/stem chordates, …). All these different views and ongoing debates about the phylogenetic position of vetulicolians should appear somewhere in such a review paper. This is not the case in the present MS.
In this context, we understand that the author considers vetulicolians as deuterostomes, but it is not clear why only these two groups/lineages of deuterostomes (vetulicolians and chordates) would be « of particular interest for understanding the evolutionary dynamics of the Cambrian evolutionary event » (line 12). What about hemichordates ? echinoderms ? cambroernids ? All these groups should be taken into account to better understand the diversification of deuterostomes in Cambrian times (and what about non-deuterostome metazoans, such as e.g. arthropods, which also underwent a major diversification in Cambrian times ?). Hemichordates and even more, echinoderms, have a much better, and more abundant and diverse fossil record than chordates or vetulicolians in the Cambrian. The author should explain why he chose to concentrate only on chordates and vetulicolians. Does he consider vetulicolians as basal/stem chordates ? Why ?
A corollary is that the deuterostome phylogeny shown on Fig. 16 is both biased and meaningless, as it includes only two deuterostome « phyla » (vetulicolians and chordates): they unsurprisingly appear as sister-groups… Such a phylogeny would greatly benefit from the inclusion of cambroernids, echinoderms and hemichordates, Another problem with that phylogeny is that it seems to be entirely hand-made (it is probably new, as no literature is cited in the figure caption), as no data matrix, no list of characters are provided anywhere. The scientific value of this phylogeny is thus very low, as there is no way of knowing how it was built. Names of taxa should be also written in larger characters (too small !!!).
Additional comments :
- lines 47-51: these sentences should be somewhat re-formulated, as they sound a bit too personal against the scientific opinion(s)/interpretation(s) exposed by Derek Briggs and Richard Fortey (and indeed many other workers, see e.g. Budd, G. 2003. The Cambrian fossil record and the origin of the phyla. Integrative and Comparative Biology, 43, 157-165). Moreover, the way this paragraph is written gives the impression that some scientists are right and others, wrong. Indeed, we are dealing here with interpretations, and science. What, if anything is a phylum (or a class, an order ?)? Nothing else than an arbitrary ranking. What is the precise phylogenetic position of vetulicolians ? This is not clearly stated anywhere in the MS. Are they basal deuterostomes ? Basal chordates ? Sister-group of craniates ? Anywhere they fit in the metazoan tree, vetulicolians (if monophyletic) are necessarily the sister-group of other animals. As many other (early-middle) Cambrian taxa, they very likely belong to the stem-group of one or several extant ‘phylum’/’phyla’. As such, the systematic rank they are assigned (phylum, class, order or even simple family) has less importance (and significance) than the position (e.g. Vetulicolia is assigned to class-level in Vinther et al. 2011 = publication [49]), where they fit in metazoan phylogeny. I would tend to concur with that statement by Aldridge et al. (2007) (= publication n°47) : « Although they [=the vetulicolians] appear to form a monophyletic clade, it is premature to accord them phylum rank without resolution of their phylogenetic position » (Aldridge et al. 2007, p. 163).
- line 218: here and elsewhere in the text, expressions such as « hard-shell » (e.g. lines 218, 275) and « soft-shell » vetulicolians (e.g., lines 225, 230, 232, 255, 333) are used without any explanation of their meaning and/or functional / anatomical / systematic relevance; please explicit these terms (this is a review paper), or do not use them;
- line 272: « evidently » is possibly too much for something that is only a hypothesis (« some type of Hox-related genetic modification has evidently led to caudal extension of the vetulicolian lateral line in Skeemella »); a more cautious statement is probably better suited here;
- lines 373-387 : all this paragraph proposing putative developmental interpretations of the vetulicolian morphology is highly speculative… The same is true for the last paragraph of the conclusion (lines 408-411).
- lines 302-363 and 413-450 : the new genus of the weird-shaped vetulicolian described here is apparently based on one(???) single specimen (the holotype). Given its very irregular, flattened and distorsted outlines, what are the arguments supporting that such a fossil does not represent a largely decayed individual of an already known/more regular species of vetulicolian ? The figures (photograph, Fig. 14 and camera-lucida drawing, Fig. 15) illustrating the new taxon are too small, and do not make it possible to visualize correctly the morphological characters of that specimen. More information about the taphonomy of that specimen should be provided, so as to better support its identification as a valid taxon, rather than as a partly crushed, flattened or disarticulated individual.
(3) References
- line 38-39: the following paper could be cited here : Mallatt, J. and Chen, J.Y. (2003) Fossil sister group of craniates : predicted and found. Journal of Morphology, 258, 1-31.
- lines 54-55: about « early western interest in Chinese Proterozoic and Cambrian strata and fossils »: it it fine to cite here Grabau (1930), however, the Lower Palaeozoic stratigraphy, geology and palaeontology of Yunnan were first investigated by French geologists from Indochina, when they came in the southern part of China for the construction of the Yunnan railway. In particular, the first evidence of early Cambrian faunas in the Chengjiang area was documented by Mansuy (1907) [Mansuy, H. (1907) Résultats de la mission géologique et minière du Yunnan méridionale. III : Résultats paléontologiques. Annales des Mines, 11, 447-471.];
- line 77: in the sentence « Assigned to the Cephalochordata [16] Cathaymyrus … » the cited reference (16) corresponds to Shu et al. (1999: Lower Cambrian vertebrates from south China. Nature, 402, 42-46), which primarily aims at describing the two (new) genera Myllokunmingia and Haikouichthys… It would be probably more correct to cite, instead of that paper, the one published by Shu et al. (1996: A Pikaia-like chordate from the Lower Cambrian of China. Nature, 384, 157-158), in which the genus Cathaymyrus is described and indeed assigned to the cephalochordates;
- line 90: ref. 16 (see above ; Shu et al., 1999) could be also cited here;
- line 93: in the paragraph on Haikouella, it is stated that ‘This early chordate and possible agnathan [20]…’ ; however, Haikouella is even not mentioned in publication n*20 [Shu et al. (2003). Head and backbone of the Early Cambrian vertebrate Haikouichthys. Nature, 421, 526-529]. The reference that should be cited here is probably [21], as it corresponds to the original description of Haikouella by Chen et al. (1999. An Early Cambrian-like chordate. Nature, 402, 518-522);
- line 94: it is mentioned that Haikouella « is represented by two species, Haikouella lanceolata and H. jianshanensis [21] »; in this sentence, it is correct to cite publication n°21 (= original description of H. lanceolata by Chen et al., 1999), but publication n°23 should be cited too (=description of the second species of Haikouella, H. jianshanensis by Shu et al., 2003);
- line 99: publications [23-26] are cited here for the first time, but publication [22] is cited only on line 164 (!!!), i.e. in between the first citations of publications [40] (line 142) and [41] (line 170)… Numbers for cited papers should be given accordingly to their first mention in the text; please correct here and everywhere else this is necessary;
Finally, the literature cited in this paper is relatively exhaustive on the topic. However, I was a bit surprised not to see any reference of publications such as :
- Butterfield, N.J. (2003) Exceptional fossil preservation and the Cambrian explosion. Integrative and Comparative Biology, 43, 166-177 : an important paper questioning the interpretation of vetulicolians as deuterostomes ;
- Cong, P.Y., Hou, X.G., Aldridge, R.J., Purnell, M.A. & Li, Y.Z. (2015) New data on the palaeobiology of the enigmatic yunnanozoans from the Chengjiang Biota, Lower Cambrian, China. Palaeontology, 58, 45-70 : a nice critical review of anatomical interpretations of yunnanozoans, surprisingly not cited here ;
- Dzik, J. (1995) Yunnanozoon and the ancestry of chordates. Acta Palaeontologica Polonica, 40, 341-360 : should be probably cited in the paragraph on Yunnanozoon ;
- Han, J., Conway Morris, S., Ou, Q., Shu, D. & Huang, H. (2017) Meiofaunal deuterostomes from the basal Cambrian of Shaanxi (China). Nature, 542, 228-231 : a very strange, basal Cambrian fossil from China (Saccorhytus coronarius) is described as a tiny deuterostome in that paper and compared with vetulicolians; it would be interesting to know what the author thinks about this fossil (this paper should be anyway mentioned, whether S. coronarius is, or not, actually a deuterostome or a vetulicolian-like fossil) ;
- Holland, N.D. & Chen, J.Y. (2001) Origin and early evolution of the vertebrates : new insights from advances in molecular biology, anatomy, and palaeontology. BioEssays, 23, 142-151. This is an interesting review on early deuterostomes (including discussions on Cathaymyrus, Haikouella, Haikouichthys, Myllokunmingia, Pikaia and Yunnanozoon) and a phylogeny;
- Li, Y.J., Cong, P.Y., Zhao, J. & Hou, X.G. (2015) New observations on morphological variation of genus Vetulicola with quadrate carapace from the Cambrian Chengjiang and Guanshan biotas, South China. Palaeoworld, 24, 36-45 : should be cited…
- Ou, Q., Conway Morris, S., Han, J., Zhang, Z., Liu, J., Chen, A., Zhang, X. & Shu, D. (2012) Evidence for gill slits and a pharynx in Cambrian vetulicolians : implications for the early evolution of deuterostomes. BioMed Central, Biology, 10:81. A recent paper on vetulicolians and their anatomy, surprisingly not cited ;
- Shu, D., Conway Morris, S., Zhang X.L., Chen, L., Li, Y. & Han, J. (1999) A pipiscid-like fossil from the Lower Cambrian of south China. Nature, 400, 746-749 : original description of the vetulicolian Xidazoon…
… and possibly also some papers on cambroernids (e.g. Hou et al., 2006, GFF; Kimmig et al. 2018, Geological Magazine), and/or early putative hemichordates (e.g. Caron et al., 2013, Nature; Hu et al. 2018, Journal of Paleontology) and/or weird basal Cambrian (?)deuterostomes, such as Yanjiahella (Topper et al., 2019; Nature Communications), which is certainly not an echinoderm, but shares with vetulicolians the possession of a longitudinal gut in its posterior appendage, with an (?)anus opening at the posterior extremity of it.
In summary, this MS would certainly benefit from some relatively substantial (moderate) revisions.
Author Response
Reply Reviewer #2
Typos fixed, thank you.
Objective now clearly stated in abstract.
Discussion of alternate interpretations of vetulicolians added
Stylophoran discussion added to revision
Enhanced discussion of vetulicolian affinities now in place with additional references.
I discuss here the possibility that vetulicolians are stem chordates.
Former Fig. 18 has been removed and the affinities discussion has been moved to the text.
Discussion of Gould, Briggs etc. has been improved.
I provide now a definition of hard shell versus soft shell vetulicolians. Discussion of Skeemella has been improved.
I have now flagged the developmental interpretations as speculative, but have
retained them in the revision.
Suggested references have been added
Detailed discussion of cambroernids, putative hemichordates and Yanjiahella are beyond the scope of the present manuscript, but I plan to consider them in future work.
Round 2
Reviewer 2 Report
Review of MS geosciences-546300-R1
« Cambrian Chordates and Vetulicolians »
submitted for publication in Geosciences
by A.S. McMenamin
revised version
Thank you for having offered me the possibility to review the revised version of this thorough review on Cambrian chordates and vetulicolians. This new version of the MS is significantly improved, compared to the original one, and the author has obviously significantly modified his text, so as to take into account the comments made during the first round of review. As it now stands, the MS offers a very-well written, up-to-date review of the current knowledge on these two groups of early deuterostomes, based on a relatively exhaustive literature, abundant illustrations, and the description of a new, weird-shaped vetulicolian from China. The diversity of interpretations about these early deuterostomes (and in particular, the vetulicolians) is much better expressed in this revised version.
While going through this revised version, I have noted only a few typos left, as well as a few problems with calls for references. My only serious concern is the absence of a formal diagnosis for the description of Shenxianyuloma nov. gen. nov. sp. So as to follow the requirements of the international code of zoological nomenclature, a short diagnosis should be added for the new genus in the systematic section (not necessary for the species, as it is the same as for the genus, by monotypy).
In summary, this revised version of the MS seems to me now acceptable for publication, if a diagnosis is added for the description of the new taxon, and if the few (very minor) points listed below are checked/modified.
List of additional (minor) suggested changes :
- line 41: please modify « vindicated ]1] » into vindicated [1] » ; is it necessary to cite twice the same paper (i.e. [1]) in the same sentence ? For the second citation, the paper currently listed as [5] (= Mallatt and Chen, 2003) might fit better here (it then should appear as [2]);
- lines 50-51 : the same word ‘credence’ is used twice in these two lines; one of them could be replaced by another word of similar meaning;
- line 51: references [4] and [5] are cited here to illustrate the « recognition of the new early Cambrian phylum Vetulicolia »; this is fine for ref. [4] (= Shu, 2008), but not for ref. [5] (= Mallatt and Chen, 2003) which is focusing on Haikouella. Please modify (as said above, the best place to cite ref. [5] would be on line 41);
- line 87 : please delete one space between « Phlogites and » and « Herpetogaster » ;
- line 220: please modify « Soft shell vetulicolians lack (…) in its posterior » into « Soft shell vetulicolians lack (…) in their posterior » [plur.];
- line 274: please modify « the line were one would expect » into « the line where one would expect »;
- line 305 : please delete one space between « vetulicolians, » and « occurs »;
- line 311: please modify « fluorence » into « fluorescence »;
- line 320: please put « Vetulicola » in italics;
- line 355: please modify « Figs. 14-22 » into « Figs. 14-23 »;
- line 474: please delete one space between « somitogenesis. » and « Interestingly »;
- line 481: please check if the paper cited here should be actually [51] (= Shu et al., 2001), or if it should not rather be [61] (Blair and Blair Hedges, 2005) ; it would make more sense;
- line 491: paper [61] is cited here for the first time, i.e. after paper [62] (first cited, line 490); please correct refs. numbers, unless there is a confusion between papers [51] and [61] on line 474 (see above);
- line 520: please modify « Soft shell vetulicolians lack (…) in its posterior » into « Soft shell vetulicolians lack (…) in their posterior » [plur.];
- line 537: please modify « of some short » into « of some sort »;
- line 543: please modify « phylum rand » into « phylum rank »;
- line 560: please add a diagnosis (see above) for Shenxianyuloma nov. gen.;
- line 628: « and two anonymous reviewers » : this is surprising and at least partly incorrect, as I signed the review I made for the original version of that MS.
Author Response
Line 41: this should read 'one hundred and eleven years' not 'ref [1]'. Reference [2] should stay as is.
second use of 'credence' changed to 'weight'
extra spaces removed
'its' changed to 'their' (both instances)
'were' change to 'where'
'fluorescense' typo fixed
Vetulicola changed to italics
ref. 51 changed to 61
'short' changed to 'sort'
diagnosis added for Shenxianyuloma